# Identifying obstacles preventing the uptake of tunnel handling methods for laboratory mice: An international thematic survey

**Lindsay J. Henderson**[1,2], **Tom V. Smulders**[1,2], **Johnny V. Roughan**[1,2]*

**1** Centre for Behaviour and Evolution, Newcastle University, Newcastle upon Tyne, United Kingdom,
**2** Institute of Neuroscience, Newcastle University, Newcastle upon Tyne, United Kingdom

* Lindsay.Henderson@ncl.ac.uk

## Abstract

Handling of laboratory mice is essential for experiments and husbandry, but handling can increase anxiety in mice, compromising their welfare and potentially reducing replicability between studies. The use of non-aversive handling (e.g., tunnel handling or cupping), rather than the standard method of picking mice up by the tail, has been shown to enhance interaction with a handler, reduce anxiety-like behaviours, and increase exploration and performance in standard behavioural tests. Despite this, some labs continue to use tail handling for routine husbandry, and the extent to which non-aversive methods are being used is currently unknown. Here we conducted an international online survey targeting individuals that work with and/or conduct research using laboratory mice. The survey aimed to identify the handling methods currently being used, and to determine common obstacles that may be preventing the wider uptake of non-aversive handling. We also surveyed opinions concerning the current data in support of non-aversive handling for mouse welfare and scientific outcomes. 390 complete responses were received and analysed quantitatively and thematically. We found that 35% report using tail handling only, and 43% use a combination of tail and non-aversive methods. 18% of respondents reported exclusively using non-aversive methods. The vast majority of participants were convinced that non-aversive handling improves animal welfare and scientific outcomes. However, the survey indicated that researchers were significantly less likely to have heard of non-aversive handling and more likely to use tail handling compared with animal care staff. Thematic analysis revealed there were concerns regarding the time required for non-aversive methods compared with tail handling, and that there was a perceived incompatibility of tunnel handling with restraint, health checks and other routine procedures. Respondents also highlighted a need for additional research into the impact of handling method that is representative of experimental protocols and physiological indicators used in the biomedical fields. This survey highlights where targeted research, outreach, training and funding may have the greatest impact on increasing uptake of non-aversive handling methods for laboratory mice.

**Data Availability Statement:** All relevant data are within the manuscript and its Supporting Information files.

**Funding:** This work was funded by a grant (NC/S000887/1) awarded to JVR and TVS by the UK National Centre for the Replacement, Refinement and Reduction of Animals in Research (NC3Rs). https://www.nc3rs.org.uk/ The funders had no role in study design, data collection and analysis, decision to publish, or preparation of the manuscript.

**Competing interests:** The authors have declared that no competing interests exist.

## Introduction

Routine handling of laboratory animals is essential for every-day husbandry and conducting experiments. However, handling can compromise the welfare of laboratory mice and is a well-recognised source of variation in animal studies [1–3]. Mice are the most widely used species in biomedical research. Consequently, evidence-based improvements to handling methods are an important refinement that could improve welfare for a large number of animals, potentially reducing the numbers required to achieve accurate and replicable findings.

The most common method used to capture and handle laboratory mice is to pick up and grasp the mouse by the base of its tail, a method often specified in standardized protocols [4–6]. However, two alternative methods for picking up laboratory mice have been investigated and validated in recent years [7–9]; 'tunnel' handling, that involves guiding mice into a tunnel before being lifted (thus avoiding direct contact), and 'cup' handling, where mice are scooped up and lifted with closed or open hands and allowed to move freely without direct physical restraint (video tutorials of techniques available online; https://www.nc3rs.org.uk/mouse-handling-video-tutorial). Both of these methods are considered non-aversive, because they result in an increased willingness of mice to interact with their handler, and have been found to lower anxiety-like behaviour and enhance the performance of mice in standard behavioural tests [7–12]. These findings have been replicated in several laboratories with a range of mouse strains [7–12]. Where examined, the positive effects of anxiety reduction, improved exploration and interaction with a handler are similar for cup and tunnel handling methods [7,8], suggesting the benefits of using either tunnel or cup methods are similar. However, it can take longer for mice to become familiarised with cupping methods [8].

Research suggests that handling mice using a home-cage tunnel can also improve ease of handling during oral gavage compared to tail-handled mice [11]. Furthermore, tail-handled mice show decreased responsiveness to reward compared to tunnel handled mice, indicative of anhedonia and chronic stress [12]. To date evidence regarding the impact of handling method upon physiological indices is limited [10,13]. A single study has shown that handling method can influence glucose metabolism [10], and there are inconsistent results regarding the influence of handling methods upon plasma corticosterone levels from two studies [10,13]. Importantly, single or repeated restraint [7,11,14,15], lifting the tail for abdominal inspection [7] and injection [11,14,16] do not negate the beneficial effects of tunnel handling upon voluntary interaction with a handler. In addition, while most studies have employed daily handling to investigate impacts upon behaviour and physiology, recent studies have shown that weekly or fortnightly handling during cage cleaning is sufficient for mice to show positive responses to tunnel handling [14,16]. Overall, non-aversive handling, especially tunnel handling methods, appear to be an important welfare refinement.

Despite evidence that non-aversive handling can enhance welfare and potentially data quality compared to tail handling; tail handling continues to be used for routine handling in some laboratories, and the extent to which non-aversive methods are being routinely used is unknown. Furthermore, the obstacles that may be preventing the wider uptake of non-aversive handling have not been formally assessed or quantified. Here, we conducted an international online survey to identify the handling methods currently being used to pick up laboratory mice, and to understand why participants use their selected handling method. We also gauged opinion as to whether current data in support of non-aversive handling methods are convincing to the research and animal care community. Finally, we sought to identify perceived obstacles that may be preventing the uptake of non-aversive handling methods.

## Methods

### Participants

The target population for this survey were individuals that work with and/or conduct research using laboratory mice. Participants were invited to complete the survey via a number of routes; advertisements within mailing lists and newsletters of 3Rs, animal care and welfare organizations, and professional research societies (full list provided in S1 Data). All participants that completed the survey were included in analysis, participants were not screened, and no inclusion or exclusion criteria were used. The project was approved by the Newcastle University Ethics Committee (7651/2018).

### Online survey

The anonymous online survey was hosted through Jisc (www.jisc.ac.uk; full survey in S2 Data, open between 15[th] September 2018–14[th] January 2019). Firstly, participant information was collected (8 questions); questions included where they had heard about the survey, their job role, place of work, time spent handling mice in the last year, how long they had worked with mice, the country they work in, and their gender and age. We then asked about their knowledge of non-aversive handling methods, and the methods they currently use to pick up mice (3 questions). Next, we asked their reasons for using their chosen handling method. In this case participants had a list of options and an opportunity to explain their choice(s) via free-form text. We then gave participants the opportunity to read summarized information on, and follow links to literature regarding the influence of handling methods upon mouse welfare and experimental outcomes (https://www.nc3rs.org.uk/how-to-pick-up-a-mouse), and asked them to rate how convinced they were by these data from 1–5, in regard to mouse welfare, and experimental outcomes (1 –not convinced, 2—not very convinced, 3 –no opinion, 4 –mildly convinced, 5 – very convinced). Finally, for thematic analysis we asked participants to explain, in a free-form response, what would be required for them to consider using tunnel handling exclusively. Throughout the survey our questions stated tunnel handling rather than using the term non-aversive handling. This approach was used to improve the specificity of the responses, however participants were always given the option to describe the use of, and reasons for using alternative non-aversive methods, such as cup handling.

### Data processing and analysis

Participants were provided with a list of options for both job role, the handling method(s) used, and the reasons for using or not using a given handling method. The majority of respondents chose options from the provided list (Job role: 83%, Handling method: 98%). But participants also had the opportunity to select an "Other" option and use free-text to describe their job role or preferred handling method. In that case, if their response was aligned with the options listed, they were coded as such. In some cases, a new category was created to account for these responses. For example, Regulator in job role, where an individual works for an organization that legislates the use of animals in scientific research.

Job role could influence both the use of, and views about, non-aversive handling methods. This is because job role could influence how much time individuals spend handling mice and/ or the type of procedures they conduct. For example, animal care staff are likely to carry out more animal husbandry than researchers. To investigate whether job role influenced the handling methods used and the opinions of the different handling methods, job roles were combined into four categories; 'Animal Care Staff', that included technicians and managers, 'Researchers', that included principal investigators, research assistants, postdocs and students,

'Veterinarians', and 'Other', that included Regulators, Teacher/Instructors and Faculty head/Directors. Where appropriate job roles were used for analysis and descriptive statistics.

Additional handling methods were added to the initial options, which were tail, cup and/or tunnel. For example, "Scruff", which is lifting the mouse out of the cage by the loose skin at the nape of the neck, "Tail-hand", which is lifting the mouse using the tail then immediately placing the mouse on the hand and "Enrichment", using another enrichment item already in the cage (e.g. a plastic "igloo") to lift the mouse. When participants were asked to give their reasons for using their chosen handling method, more options were available to explain why they do not use tunnel handling (N = 16), compared with why they did use non-aversive methods (N = 5). For both questions, participants were given the opportunity to give additional reasons in free-form text. Based on the responses in the free-form text, we created additional reasons for using handling methods. As participants could choose more than one reason for using the chosen handling methods, percentages can sum to more than 100. Responses to the multiple-choice questions were analysed quantitatively using Pearson's chi-squared tests and descriptive statistics (means/percentages) in R 3.3.2 (R Development Core Team, 2016).

Not all respondents provided qualitative comments for the optional question, "What would be required for you to consider using tunnel handling routinely?", that offered a free-form response (N = 249). As this question specifically targeted respondents that were not routinely using tunnel handling, responses were gathered from a sub-sample of respondents. Responses to this question were analysed thematically and coded according to the main themes they described. Each comment was initially coded separately, then individual comments were re-grouped into categories that were similar or equivalent in theme. Some of the original coded comments were grouped based on topical overlap of comments (e.g., grouped within "*Perceived practicality of tunnel handling*" were comments regarding using tunnels for procedures and health checks, use of tunnels in specific cage sizes, and for experimental apparatus). The qualitative comments were initially coded by one author (LJH) and verified by the other authors (TVS, JVR). A response that addressed multiple themes was counted as multiple comments, therefore percentages reported can sum to more than 100.

## Results

### Participants

The demographic characteristics of the 390 participants who completed the survey are detailed in Table 1. Briefly, respondents worked in 27 countries, and the majority were female (72%). Respondents included researchers, animal care professionals, veterinarians, regulators of the use of animals in scientific research, and animal health and welfare professionals (see Table 2a and 2b for full details). The majority of participants were animal care professionals (N = 148 (38%)) and researchers (N = 192 (49%)), making up 87% of respondents. Participants varied in the frequency with which they routinely undertook mouse handling, with animal care staff handling mice most often (see Fig 1a). At the time of the survey, most participants had over 10 years of experience handling laboratory mice (see Fig 1b).

### Knowledge and use of handling methods

The majority of participants stated they had prior knowledge of tunnel handling methods before completing the survey (mean across countries; 80.9%). At the time of the survey, participants reported using a range of handling methods for picking up laboratory mice (see Table 3). Respondents that used a combination of tail and non-aversive handling methods were the largest group represented with 43%. 35% of respondents reported using only tail methods to pick up mice, and 18% of respondents used only non-aversive handling methods

**Table 1. Summary of the demographic information of survey participants (o = optional questions, m = mandatory questions, N = number of responses).**

| Question | Answer options | Count | % of N | N |
|---|---|---|---|---|
| *Age (o)* | 16–25 | 39 | 10.1 | 386 |
| | 26–35 | 141 | 36.5 | |
| | 36–45 | 109 | 28.2 | |
| | 46–55 | 65 | 16.8 | |
| | 56–65 | 31 | 8.0 | |
| | >65 | 1 | 0.3 | |
| *Gender (o)* | Female | 272 | 71.6 | 380 |
| | Male | 108 | 28.4 | |
| *Country (m)* | Australia | 35 | 9.0 | 390 |
| | Austria | 1 | 0.3 | |
| | Brazil | 2 | 0.5 | |
| | Canada | 54 | 13.8 | |
| | Chile | 1 | 0.3 | |
| | Czech Republic | 1 | 0.3 | |
| | Denmark | 1 | 0.3 | |
| | Finland | 1 | 0.3 | |
| | France | 20 | 5.1 | |
| | Germany | 23 | 5.9 | |
| | Hungary | 2 | 0.5 | |
| | India | 1 | 0.3 | |
| | Ireland | 6 | 1.5 | |
| | Italy | 2 | 0.5 | |
| | Latvia | 1 | 0.3 | |
| | Lithuania | 4 | 1.0 | |
| | México | 1 | 0.3 | |
| | Netherlands | 5 | 1.3 | |
| | New Zealand | 29 | 7.4 | |
| | Portugal | 6 | 1.5 | |
| | South Africa | 1 | 0.3 | |
| | Spain | 2 | 0.5 | |
| | Sweden | 7 | 1.8 | |
| | Switzerland | 33 | 8.5 | |
| | Thailand | 1 | 0.3 | |
| | UK | 119 | 30.5 | |
| | USA | 31 | 7.9 | |

(see Table 3). The handling methods used were similar across the main job roles, however researchers used tail only methods significantly more than the other handling methods compared with animal care staff (Pearson's chi-squared: $\chi^2 = 7.54$, P < 0.01, Fig 2).

To understand why participants preferred a particular handling method they were asked to select reasons for their choice. The main reasons respondents gave for using tunnel handling were "Benefits to animal welfare", "Benefits to experimental outcomes" and "Guidelines at place of work", with 88%, 44% and 35% choosing those options respectively (see Table 4). There was a wider range of reasons chosen for not using tunnel handling methods. The top three responses were, "I use the handling methods that have always been used", "I am concerned it will be slower" and "Not sure it's better than current method", with 32%, 31% and

**Table 2. Breakdown of participants by A) job role, and B) place of work (m = mandatory question, N = number of responses).**

| A) | | | | |
| --- | --- | --- | --- | --- |
| Question | Answer options | Count | % of N | N |
| Job role (m) | Animal care manager | 45 | 11.5 | 390 |
| | Animal care technician | 103 | 26.4 | |
| | Faculty head/ Director | 3 | 0.8 | |
| | Instructor/Teacher | 2 | 0.5 | |
| | Researcher—Technician/Assistant | 25 | 6.4 | |
| | Researcher–Post doc/ Associate | 49 | 12.6 | |
| | Researcher–Principal investigator | 67 | 17.2 | |
| | Researcher–Student | 51 | 13.1 | |
| | Regulator | 16 | 4.1 | |
| | Veterinarian | 29 | 7.4 | |
| B) | | | | |
| Place of work (m) | Non-governmental oversight body | 2 | 0.5 | 390 |
| | Private Company R&D | 35 | 9.0 | |
| | Publicly Funded Research Institute | 76 | 19.5 | |
| | Research Hospital | 5 | 1.3 | |
| | University/ College | 270 | 69.2 | |
| | Other | 2 | 0.5 | |

30% of respondents choosing these options respectively. Also, 26% of respondents chose "I had not previously heard of tunnel handling" and 24% chose "No one has suggested to do it differently". 22% of respondents chose the "Other" option, the most common reason given in this case was the perceived incompatibility of tunnel handling with restraining mice for procedures and health checks (13%). The second most common response was that they prefer cup handling (5%) (see Table 5 for all responses categorised by "Other").

When we compared the reasons chosen for not using tunnel handling between animal care staff (N = 107) and researchers (N = 152), both groups selected "I am concerned it will be slower" and "I use the handling methods that have always been used" in their top three responses (Table 6). However, animal care staff selected "Not sure it's better than current method" in their top three responses, while researchers selected "I had not previously heard of tunnel handling" (Table 6). "I had not previously heard of tunnel handling" was chosen significantly more by researchers compared with animal care staff (14% difference, Pearson's chi-squared: $\chi^2$ = 5.90, P = 0.01, Table 6). The other reasons that differed by more than 5% between the job roles were, "No one has suggested to do it differently", "I use the handling methods that have always been used" and "Experimental continuity" which were chosen 8%, 7% and 6% more respectively by researchers compared with animal care staff but these did not differ significantly (Table 6). "Financial considerations; purchase of tunnels" was selected 5% more by animal care staff compared to researchers (Table 6). However, these differences were not statistically significant (Pearson's chi-squared: P > 0.20).

## Views on the impact of tunnel handling upon mouse welfare and scientific outcomes

After being given the opportunity to review a summary of the evidence describing the effects of tunnel handling upon mouse welfare and scientific outcomes, the most common responses of participants were that they were either mildly or very convinced that tunnel handling improved mouse welfare and scientific outcomes (Fig 3a and 3b). For both animal care staff

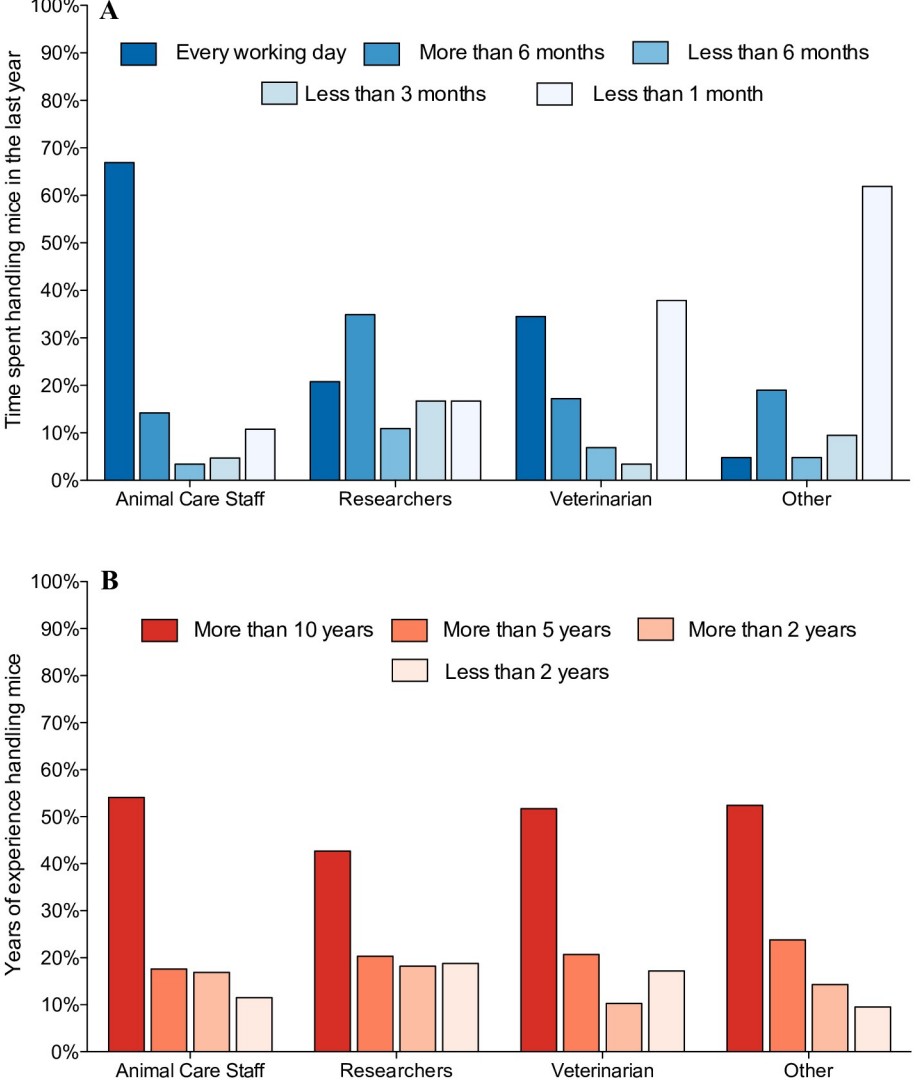

**Fig 1.** A) Percentage of respondents that had spent, either every working day, > 6 months, < 6 months, < 3 months or < 1 month handling mice in the past year, by job role. B) Percentage of respondents that had worked with mice for > 10, > 5 or > 2 years, or < 2 years, by job role, (N = 390). "Other" job role category includes Faculty head/ Directors, Regulators and Teacher/ Instructors, and constitutes 5% of respondents.

and researchers, 12% more choose "very convinced" for welfare compared with scientific outcomes. Whereas, 13% more animal care staff choose "no opinion" for scientific outcomes compared with welfare (Fig 3a and 3b). The "not convinced" and "not very convinced" categories differed by less than 2% between scientific outcomes and welfare for both animal care staff and researchers.

## Thematic outcomes

To identify obstacles preventing the uptake of tunnel handling methods, we asked respondents what would be required for them to consider using tunnel handling routinely. As this question specifically targeted respondents that were not routinely using tunnel handling, we expected responses from a sub-sample of respondents. 249 out of 390 total survey participants provided

**Table 3. Summary of methods used by survey participants to pick up laboratory mice (N = 389).**

| Handling Method | Count | % of respondents | Count (per method) | % of respondents (per method) |
|---|---|---|---|---|
| *Tail* | 136 | **35.1** | 136 | 35.1 |
| *Non-aversive methods* | 68 | **17.5** | | |
| Tunnel, Cup | | | 37 | 9.5 |
| Tunnel | | | 22 | 5.7 |
| Cup | | | 9 | 2.3 |
| *Tail and non-aversive methods* | 168 | **43.3** | | |
| Tail, Tunnel | | | 73 | 18.8 |
| Tail, Tunnel, Cup | | | 59 | 15.2 |
| Tail, Cup | | | 34 | 8.5 |
| Tail, Enrichment | | | 1 | 0.3 |
| Tail, Tunnel, Cup, Enrichment | | | 1 | 0.3 |
| Tail, Tunnel, Cup, Enrichment, Scruff | | | 1 | 0.3 |
| *Tail and other handling methods* | 16 | **4.1** | | |
| Tail, Scruff | | | 6 | 1.5 |
| Tail, Forceps | | | 3 | 0.8 |
| Scruff | | | 2 | 0.5 |
| Tail-Hand | | | 2 | 0.8 |
| Tail, Tail-Hand | | | 2 | 0.5 |

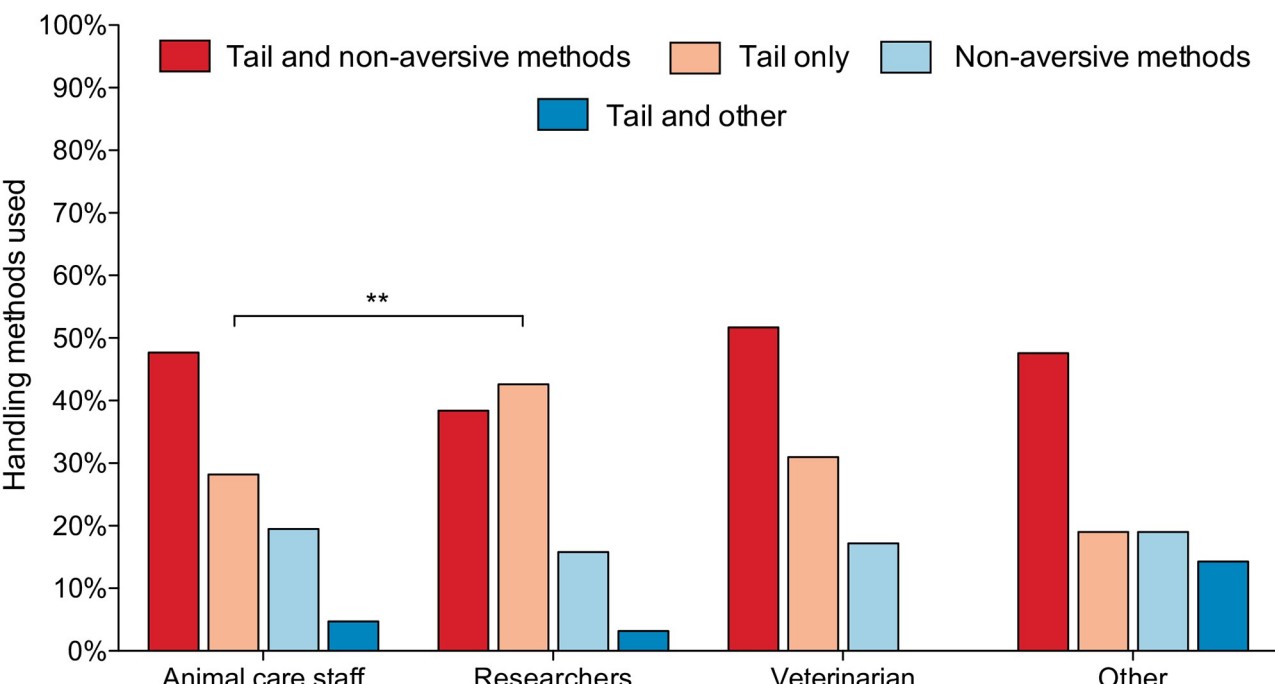

**Fig 2. Handling methods used to pick up mice by job role; animal care staff (N = 149), researchers (N = 190), veterinarian (N = 29) and other (N = 21).** "Other" job role category includes Faculty head/ Directors, Regulators and Teacher/ Instructors, and constitutes 5% of respondents. ** denotes P < 0.01. See S3 Data for a comparison of the handling methods used between the UK and the other countries represented in the survey.

**Table 4. Answer options chosen by survey participants, and elective responses provided under the "Other" option by survey participants for using tunnel handling to pick-up laboratory mice.**

|  | Answer options | Count | % of N | N |
|---|---|---|---|---|
| ***Why do you use tunnel handling?*** | Benefits to animal welfare | 169 | 87.6 | 193 |
|  | Benefits to experimental outcomes | 85 | 44.0 |  |
|  | Guidelines at place of work | 67 | 34.7 |  |
|  | **Elective responses** |  |  |  |
|  | Mouse already in tunnel | 8 | 4.1 |  |
|  | Faster and/or more efficient | 4 | 2.1 |  |
|  | Improves mouse behaviour | 3 | 1.6 |  |
|  | Trialling non-aversive methods | 2 | 1.0 |  |
|  | Use only if mouse is habituated to tunnel | 2 | 0.5 |  |
|  | Personal well-being | 1 | 1.0 |  |

free-text responses (64%), 208 of those could be coded and used in the thematic analysis (53% of total survey participants). 43% of total survey participants were potentially not routinely using tunnel handling, as they reported using a combination of tail and non-aversive methods, and 39% of total survey participants reported not using any form of non-aversive methods (82% of total survey participants, see Table 3). Therefore, a substantial proportion of respondents, that may not have been using tunnel handling routinely provided qualitative responses to the question. Percentages reported below are as a proportion of the total number of responses that could be thematically coded (N = 208).

**Table 5. Reasons given by respondents for not using tunnel handling to pick up laboratory mice that were not covered by the options provided (N = 293).** These reasons were provided under the "Other" option or taken from the explain-your-choice free-text field and thematically coded. Participants could describe multiple reasons.

| "Other" responses | Count | % of N |
|---|---|---|
| Incompatible with scruff/ health check/ procedure | 37 | 12.6 |
| Prefer cup methods | 15 | 5.1 |
| When tail handling is done quickly by experienced handler there is no benefit to using non-aversive methods | 11 | 3.8 |
| Availability of tunnels | 9 | 3.1 |
| Tunnel more stressful than tail | 8 | 2.7 |
| Biosecurity | 7 | 2.4 |
| Incompatible with experimental apparatus | 5 | 1.7 |
| Would require top down change | 4 | 1.4 |
| Researcher dependent | 3 | 1.0 |
| Use the best method at the time to pick-up mice | 3 | 1.0 |
| Habituation time | 3 | 1.0 |
| Aggressive mouse | 2 | 0.7 |
| Last resort when non-aversive methods don't work | 2 | 0.7 |
| Staff resistance to change | 2 | 0.7 |
| Currently trialling tunnel handling | 2 | 0.7 |
| Tunnel handling causes mice to become resistant to other handling | 2 | 0.7 |
| Doesn't work for specific mouse strain | 2 | 0.7 |
| Unconvinced by evidence it is better than tail handling | 2 | 0.7 |
| Breeding colony | 1 | 0.3 |
| Cage size | 1 | 0.3 |
| Continuity of teaching | 1 | 0.3 |

**Table 6. Reasons given by survey participants for not using tunnel handling to pick-up laboratory mice by the two main job roles (N = 259).** Participants could select multiple options therefore the options sum to >100%. Difference of > 5% in the reasons selected between Researchers and Animal Care Staff are highlighted in bold. * denotes P = 0.01.

| Reasons for not using tunnel handling | % Researchers | % Animal Care Staff |
|---|---|---|
| *I use the handling methods that have always been used* | **35** | **28** |
| I am concerned it will be slower | 31 | 33 |
| Not sure it's better than current method | 29 | 29 |
| *I had not previously heard of tunnel handling**  | **32** | **19** |
| *No one has suggested to do it differently* | **28** | **21** |
| *Experimental continuity* | **17** | **11** |
| *Financial considerations; purchase of tunnels* | 8 | 13 |
| Time required for retraining | 9 | 10 |
| Tunnel handling has not been validated for my experimental paradigm | 9 | 7 |
| Handling method unimportant when mice also undergo additional procedures | 9 | 6 |
| Access to retraining | 5 | 6 |
| Financial considerations; additional staff resources | 4 | 6 |
| Possible negative influence upon experimental outcomes | 4 | 5 |

Thematic coding identified issues pertaining to: perceived time constraints of tunnel handling and training (N = 59 (28% of responses)); practicality of tunnel handling for health checks, procedures and experimental testing (N = 54 (26%)); availability of tunnels, cost of tunnels, and biosecurity (N = 54 (26%)); a need for further evidence in support of using non-aversive handling methods (N = 49 (24%)); and a need for top-down implementation and consensus about the use of non-aversive handling methods (N = 40 (19%)).

## Time constraints of tunnel handling or training

Time constraints were cited by respondents as the most common reason for not using tunnel handling. While participants stated that tunnel handling may only take a few seconds more per mouse compared with tail handling, ultimately, they were concerned that this relatively small increase would result in a significant overall increase in the time required to process large numbers of cages. Respondents stated that they would have to see further proof that using tunnel handling is as fast as tail handling for cage cleaning and health checks, and/or that the benefits to welfare and experimental outcomes outweigh any increase in the time required for handling. This, however, conflicts with the opinion of other respondents that stated mice are often found already residing in the tunnel when the time comes to lift them. In this case, participants stated that tunnel handling can be faster than tail handling for picking up mice. The time required for retraining was also mentioned as an obstacle to using tunnel handling.

## Perceived practicality of tunnel handling

The second most common reason participants gave for not using tunnel handling was the perceived incompatibility of tunnel handling with health checks and common procedures that require the mouse to be restrained. Another reason respondents gave for not using tunnel handling was that they believe mice can become more stressed, aggressive or harder to handle when they have not experienced direct contact with hands. Therefore, in their opinion tunnel handling may increase the distress experienced by mice when they need to be handled by hand

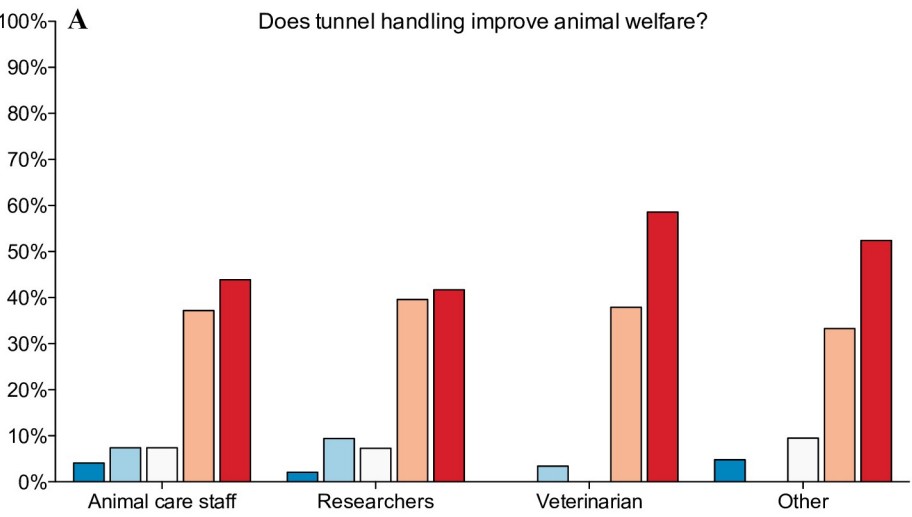

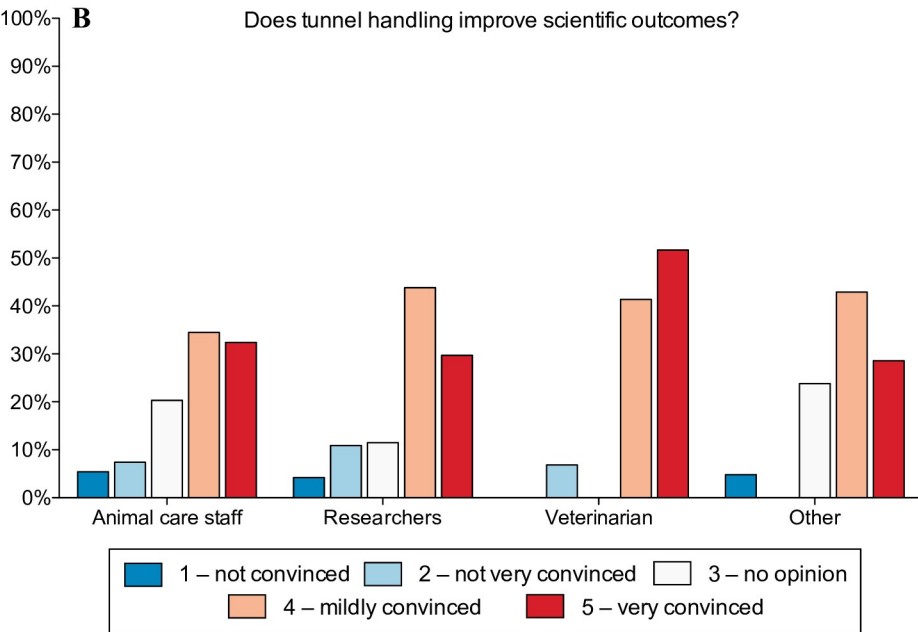

**Fig 3. Views on the impact of tunnel handling upon mouse welfare and scientific outcomes by job role.** Percentage of respondents that were; 1—not convinced, 2—not very convinced, 3—no opinion, 4—mildly convinced, 5—very convinced that, A) tunnel handling improves welfare, and B) tunnel handling improves scientific outcomes, (N = 390). "Other" category for job role includes Faculty head/ Directors, Regulators and Teacher/ Instructors, and constitutes 5% of respondents.

for restraint, procedures or experimental tests. Respondents also raised the issue of incompatibility of tunnels with experimental apparatus, cages or implants. These comments included limited space within cages for tunnels, and mice not being able to fit in tunnels due to implants. Also, the size or shape of experimental apparatus preventing the use of a tunnel to transport the mouse from the home-cage to the experimental test. Finally, some tests such as the grip-strength test require rapid lifting and replacing of a mouse upon a surface, which precludes the use of a tunnel.

### Availability of tunnels, cost and biosecurity

Another common reason respondents gave for not using tunnel handling is the availability of tunnels; either they were not available within their facility, only cardboard tunnels were available and mice would quickly destroy them causing them to not be available for handling, or there are not enough tunnels for one to be allocated to each cage. This latter point was primarily considered a potential biosecurity issue, with respondents anticipating an increased risk of contamination when sharing tunnels between cages. The main reason provided for insufficient numbers of tunnels was the cost.

### Further evidence in support of using non-aversive handling methods

Respondents highlighted a need for further evidence in support of other non-aversive handling methods, and this can be split into two issues. Firstly, that the tail handling in previous studies is not comparable to the methods used in practise. Respondents stated that if tail handling is done quickly by a well-trained handler there is unlikely to be a detrimental effect upon mouse welfare. Specifically, that studies that have compared tail and tunnel handling have held the tail for longer than would be common during routine handling, or handler experience of using a tunnel has not been taken into account. Respondents also highlighted the need to validate whether non-aversive methods for picking up mice continue to be beneficial when mice also need to undergo regular restraint for health checks or procedures, such as injections.

Secondly, participants stated that the majority of evidence in support of tunnel handling is restricted to behavioural outcomes and they would require evidence that handling method also impact physiological measures. Respondents stated that more convincing evidence would be an impact of handling method upon stress physiology, e.g. serum or plasma glucocorticoid concentrations, heart rate and blood pressure. Also, studies that investigate an effect of handling method upon cardiovascular indicators and the outcomes of surgical procedures, anaesthesia, and drug delivery. Furthermore, examining the effect of handling method within oncological research would be insightful. Researchers also stated that they would need to validate the method within their experimental paradigm, before using non-aversive methods routinely.

### Top-down implementation and consensus

Participants highlighted the need for top-down changes to handling norms. Both researchers and animal care staff would need to be convinced, and resources would need to be invested in purchasing tunnels and training. Respondents stated this would only happen consistently when change happened at an institution level or when legislation was changed.

## Discussion

This survey suggests that opinions regarding the published benefits of non-aversive mouse handling have not necessarily transitioned into routine handling practise in the laboratory. Our results indicate that most respondents were aware of non-aversive handling methods for picking up laboratory mice and thought non-aversive methods likely to be beneficial for mouse welfare and experimental outcomes. Yet, 35% of respondents reported using tail handling exclusively, and a further 43% reported using a combination of tail and non-aversive methods. Importantly, our survey did not identify the frequency with which respondents that use a combination of handling methods, used either tail or non-aversive techniques. If respondents that use a combination of handling methods restrict tail handling only to situations where non-aversive methods are unsuitable, our results would suggest that the majority of respondents (61%) regularly use non-aversive handling methods (sum of 43% using a

combination and 18% non-aversive only). However, respondents that use a combination of handling methods may employ tail handling for other reasons, for example experimental continuity for specific studies, due to perceived time differences between methods, or for health checks and procedures. In this case, this result would suggest that non-aversive handling methods are not the default method used for picking up mice. This highlights the need for a more nuanced understanding of when non-aversive handling methods are being implemented across laboratories. Ultimately, these results indicate there is a greater scope for the uptake and implementation of non-aversive handling.

The responses to this survey provided insight both into the reasons non-aversive methods are being used and the issues that may be preventing their widespread use. Predicted improvement to animal welfare was the most common reason for using non-aversive methods, with almost 90% of respondents reporting this as their main reason for using non-aversive methods. Whereas about half cited benefits to experimental outcomes, and one third specified guidelines at their place of work. These results may indicate that the benefits of non-aversive handling to scientific outcomes are either less important or less convincing as compared to welfare outcomes. Indeed, when asked to rate how convinced they were that tunnel handling improves animal welfare or scientific outcomes, 12% more participants were very convinced by welfare benefits compared with scientific outcomes. Additionally, more participants choose "no opinion" for scientific outcomes compared with welfare. The greater selection of "no opinion" for scientific outcomes may reflect concerns by participants of their ability to judge the validity of data in support of the benefits of non-aversive handling upon scientific outcomes. These results suggest further evidence of the impact of handling methods upon scientific outcomes would be valuable.

The survey highlighted a range of reasons participants do not use non-aversive methods for picking up laboratory mice. Reasons such as "I use the methods that have always been used", "Not sure it is better than the current method", and "No one suggested to do it differently" were each selected by more than a quarter of respondents. These reasons suggest a reluctance to change current handling methods. Responses to the survey also indicate that compared with animal care staff, researchers were more likely to use tail handling methods and were less likely to have heard of tunnel handling. This suggests one of the main reasons researchers do not use tunnel handling is due to being unfamiliar with the method. Indeed, when we removed participants that had not heard of tunnel handling from the analysis, there was no longer a significant difference between the number of researchers using tail handling compared to animal care staff (Pearson's chi-squared: $\chi^2$ = 2.87, P = 0.09). Therefore, targeting information toward researchers on non-aversive handling methods and obtaining evidence that changing handling methods will not impede experimental continuity will likely expand the use of tunnel handling.

The most commonly cited concern selected by both researchers and animal care staff was that tunnel handling would be slower than picking up mice by the tail; 33% in the case of care staff and 31% of researchers. This concern was raised by respondents that used a combination of tail and non-aversive methods, as well as those that use tail methods only. However, this concern was more common among respondents that use tail handling to pick up mice (32% vs. 22%). The perceived time constraints of tunnel handling compared to tail handling was also the most commonly cited issue identified in the thematic analysis. In contrast to this, some respondents stated that speed and efficiency were reasons that they use tunnel handling (Table 5). Clearly, there is conflicting opinions on the time required for tail versus non-aversive methods. Accordingly, in the thematic analysis respondents stated the need for additional research to resolve this issue, especially in studies involving larger numbers of mice. This is because for cage cleaning and some experimental designs, hundreds of mice need to be

processed in a day. To date, however, published studies that have compared standard tail-handling to non-aversive methods have not explicitly quantified the time taken to handle mice between the two methods, and the number of mice processed per day have rarely exceeded 100 (but see [7]), more commonly sample sizes were less than 50 [17]. Research comparing the time required between the handling methods has been conducted within individual facilities and establishments, but these studies are often only shared internally or presented at meetings, and not shared more broadly (pers. comm.). Overall, resolving these concerns necessitates additional published studies that quantify time scales between methods, particularly those involving larger sample sizes, e.g. in breeding facilities.

Another issue pertaining to differences in the time required between non-aversive and tail methods was biosecurity. Respondents stated that having to share tunnels between cages would mean cleaning before next use. In this case, a home-cage tunnel may be required, or cup handling may be a more appropriate non-aversive method. However, cup handling may not be suitable for mice infected with zoonotic pathogens, as respondents raised the issue of biting risk. In addition, some strains, for example C57BL/6 mice have been shown to habituate more slowly to handling via a shared tunnel, compared with a home-cage tunnel [9]. The use of home-cage tunnels may therefore be the most effective solution to these issues, and may be especially helpful in studies involving anxious strains [9]. Micro-grants that fund facilities to ensure they have sufficient tunnel numbers and that staff have appropriate training would be useful to combat this issue.

Respondents raised the issue of perceived incompatibility of tunnels with experimental apparatus, cages or implants. These comments included limited space within cages for tunnels, and mice not being able to fit in tunnels due to implants. For these situations cup handling may be a more appropriate alternative to tail handling. The positive effects upon anxiety reduction and interaction with a handler mostly generalise across strains, handlers, and the light/dark phase for both cupping and tunnel handling methods [7]. Also, mice handled by tunnel or cupping methods showed improved performance in a habituation-dishabituation test compared to mice picked up by the tail [8]. This suggests the benefits of using either tunnel or cupping methods, albeit not identical, are comparable. The uptake of tunnel handling for picking up laboratory mice has also led to a wider variety of tunnel sizes and designs being manufactured, which has the potential to solve issues of unsuitability of tunnels raised in this survey.

The thematic analysis also highlighted the concern of incompatibility of tunnel handling with common procedures, such as health checks and injections, that require a mouse to be restrained. Mice are most commonly restrained by holding the tail and then grasping the loose skin of the nape of the neck, a procedure commonly referred to as 'scruffing'. From the responses in the free-form text, respondents often pick up the mouse by the tail from the cage for restraint. It was unclear if respondents also consider holding the mouse in place by the tail for restraint "tail handling", rather than specifically picking the mouse up by the tail. Nevertheless, mice do not need to be picked up by the tail for restraint, abdominal inspection or procedures. There is evidence from multiple mouse strains that single or repeated restraint, where the mouse is held in place by the tail but not lifted by the tail, does not negate the benefits of tunnel handling upon voluntary interaction with a handler [7,14,16]. Raising the back end of the mouse using the tail for abdominal inspection was also not aversive if mice were picked up and placed on the hand by tunnel or cupping methods [7]. Thus, reduced interaction with a handler and greater anxiety in behavioural tests caused by tail handling are likely due to being captured and picked up by the tail, rather than the tail being manipulated *per se*.

A further concern was that repeated procedures and restraint may negate any benefits of non-aversive handling methods. However, current evidence does not support this. In ICR mice, tunnel handling increased voluntary interaction with a handler compared with tail

handling, even after a week of daily restraint and oral gavage of saline [11]. This study also provided evidence that tunnel handled mice showed greater exploration in an Open Field Test (OFT) and Elevated Plus Maze (EPM) following a single IP injection in comparison with tail handled mice [11]. In addition, repeated injection [14,16], tattooing or ear-tagging [15], do not negate the beneficial effects of tunnel handling upon voluntary interaction with a handler. A concern identified in the thematic analysis was that respondents believed mice can become more stressed, aggressive or harder to handle after tunnel handling because they have not experienced direct contact with hands. Yet, Nakamura et al. provide evidence that tunnel handling improved ease of handling (rating scale for wildness [18]) during oral gavage compared to tail handled mice [11]. Also, the duration of handling by tail or tunnel methods (2–60 s) does not influence the beneficial effects of tunnel handling, and tunnel handling for as little as four fortnightly cage cleans, can substantially increase voluntary interaction with a handler compared with tail handled mice [14]. These results further suggest that it is picking the mouse up by the tail rather than restraint or undergoing a procedure, that increases anxiety and aversion to a handler. Furthermore, brief handling is sufficient for mice to show positive responses to tunnel handling [14,16]

Survey respondents perceived the majority of evidence in support of non-aversive handling to be restricted to behavioural outcomes and stated further evidence that handling methods also impact physiological indices would increase their likelihood of using non-aversive methods. To date a couple of studies have investigated the impact of handling method upon physiological measures [10,13]. In a study that compared tail and cup methods, mice picked up by the cupping method showed a reduction in blood glucose levels, in addition to reduced anxiety-like behaviours in the EPM, compared to mice handled by the tail [10]. Furthermore, cup handled mice maintained on a high-fat diet for three months exhibited improved glucose tolerance compared to tail-handled controls [10]. The impact of handling methods on the severity of symptoms in the ICGN glomerulonephritis mouse (a model for the human idiopathic nephrotic syndrome) has also been examined [13]. In female mice histopathological scores of glomerulus lesions were significantly higher for tail handled mice compared with control mice that did not receive a protocol of daily handling [13]. Glomerulus lesion scores for tunnel and cup handled mice were intermediate between control and tail handled mice, but they did not significantly differ from control or tail groups [13]. However, a small sample size was used in this study (N = 5 per sex for each handling group; control, tail, cup, tunnel), therefore the statistical power may not have been sufficient to detect significant effects.

The impact of handling methods upon response to reward has also been investigated. Tail handled mice showed decreased responsiveness to a sucrose reward compared to tunnel handled mice, indicative of anhedonia and a depressive-like state [12]. The impact of tail handling upon response to reward is similar to that caused by chronic stress protocols used to create models of depression in mice [12,19]. Such manipulations are well-known to have neural and physiological effects on the mice [20]. Taken together these studies provide evidence that handling method can influence indices of chronic stress, impact physiology and potentially alter disease progression. But as yet these studies are limited in number and scope. Specifically, studies that have addressed the influence of handling methods upon circulating corticosterone in laboratory mice are limited. A single study has shown that tail handled mice had higher plasma corticosterone than tunnel handled mice after exploration of an EPM [10]. Ono et al. (2016) show that corticosterone was strain dependent; in C57BL/6 mice, plasma corticosterone (measured 20 mins after handling) was significantly higher in tail handled mice compared with unhandled controls but did not differ between tunnel and tail handled mice. Whereas, in BALB/c mice, plasma corticosterone was significantly higher in tunnel handled mice compared with tail handed mice and controls. However, this result was confounded by handling duration; handling duration was

longer for tunnel handled BALB/c mice as they took longer to voluntarily enter the tunnel compared with C57BL/6 mice. Importantly, allowing mice to voluntarily enter a tunnel is also not the recommended practise, rather it is recommended that mice are guided into a tunnel [7]. Therefore, additional research that investigates the influence of handling method upon stress physiology is required. Furthermore, future research that investigates the importance of handling methods upon tumour growth, cardiovascular indicators, and the outcomes of surgical procedures, anaesthesia and drug delivery, would improve our understanding of how handling methods influence scientific outcomes.

This survey aimed to identify obstacles that may be preventing the uptake of non-aversive handling methods. Concerns including lack of resources, perceived practicality and time constraints of non-aversive handling were highlighted. The growing evidence base provides a general consensus on the benefits of non-aversive handling upon mouse welfare. However, respondents highlighted a need for further studies that are representative of real-life scenarios in biomedical research. Overall, additional research, and targeted outreach, training and funding could have a substantial impact upon increasing the uptake of non-aversive handling methods for laboratory mice.

## Recommendations

- Our results suggest researchers are less likely to have heard of non-aversive handling methods compared with animal care staff. Therefore, targeting information delivery to researchers may improve the uptake of non-aversive handling.

- This survey suggests that non-aversive methods were exclusively used for picking up laboratory mice by fewer than 20% of respondents. However, this may underrepresent the number of practitioners routinely using non-aversive methods, as 35% reported using a combination of tail and non-aversive methods. Understanding the frequency of tail handling by individuals that report using a combination of handling methods would provide a more accurate insight into the uptake of non-aversive methods.

- The issue of tunnel availability and training may be resolved by micro-grants that fund facilities to ensure they have sufficient tunnel numbers and that staff have appropriate training.

- Concerns that non-aversive methods may be slower, are incompatible with restraint and common procedures, and have not been explored within a number of biomedical disciplines, would be addressed by additional published studies that address these knowledge gaps. Specifically, direct research into the effects of handling methods upon physiological outcomes that are relevant to biomedical research (stress physiology, cardiovascular indicators and oncology).

## Supporting information

**S1 Data. List of organisations and mailing lists.**
(DOCX)

**S2 Data. Laboratory mouse handling.**
(PDF)

**S3 Data.**
(DOCX)

**S4 Data.**
(XLSX)

## Acknowledgments

The study was approved by the ethics committee of Newcastle University, and all methods were carried out in accordance with the approved guidelines. We would like to thank Jasmine Clarkson, Jane Hurst and Candy Rowe for their comments on the manuscript.

## Author Contributions

**Conceptualization:** Tom V. Smulders, Johnny V. Roughan.

**Data curation:** Lindsay J. Henderson.

**Formal analysis:** Lindsay J. Henderson.

**Funding acquisition:** Tom V. Smulders, Johnny V. Roughan.

**Investigation:** Lindsay J. Henderson.

**Supervision:** Tom V. Smulders.

**Writing – original draft:** Lindsay J. Henderson.

**Writing – review & editing:** Lindsay J. Henderson, Tom V. Smulders, Johnny V. Roughan.

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
