## [Decision Letter · Decision Letter 0]

5 Feb 2020

PONE-D-19-32020

Identifying obstacles preventing the uptake of tunnel handling methods for laboratory mice: An international thematic survey

PLOS ONE

Dear Dr. Henderson,

Thank you for submitting your manuscript to PLOS ONE. After careful consideration, we feel that it has merit but does not fully meet PLOS ONE’s publication criteria as it currently stands. Therefore, we invite you to submit a revised version of the manuscript that addresses the points raised during the review process.

We would appreciate receiving your revised manuscript by Mar 21 2020 11:59PM. To enhance the reproducibility of your results, we recommend that if applicable you deposit your laboratory protocols in protocols.io, where a protocol can be assigned its own identifier (DOI) such that it can be cited independently in the future. For instructions see: http://journals.plos.org/plosone/s/submission-guidelines#loc-laboratory-protocols

We look forward to receiving your revised manuscript.

Kind regards,

Kathleen R. Pritchett-Corning, D.V.M.

Academic Editor

PLOS ONE

2. Your ethics statement must appear in the Methods section of your manuscript. If your ethics statement is written in any section besides the Methods, please move it to the Methods section and delete it from any other section. Please also ensure that your ethics statement is included in your manuscript, as the ethics section of your online submission will not be published alongside your manuscript.

Additional Editor Comments (if provided):

My apologies for the delay on this. The comments are minor, I feel, and should be easily addressed. I promise greater alacrity in the return of a revised MS.

Reviewers' comments:

Reviewer's Responses to Questions

**Comments to the Author**

1. Is the manuscript technically sound, and do the data support the conclusions?

Reviewer #1: Yes

Reviewer #2: Partly

2. Has the statistical analysis been performed appropriately and rigorously? 

Reviewer #1: Yes

Reviewer #2: N/A

3. Have the authors made all data underlying the findings in their manuscript fully available?

Reviewer #1: Yes

Reviewer #2: Yes

4. Is the manuscript presented in an intelligible fashion and written in standard English?

Reviewer #1: Yes

Reviewer #2: Yes

5. Review Comments to the Author

Reviewer #1: Overall

Solid manuscript and experiment overall. Well written, well designed, and timely. My comments below are written with the line number they reference proceeding.

Abstract

14 – If there is space include that handling methods can also influence physiological data. Also the sentence would be better written as a list such as “influence behavior, potentially reduce replicability…, and compromise mouse welfare.” The fact that the method used influences performance in behavioral test is not the reason that mouse welfare is compromised.

16 – If space, include brief definition of non-aversive handling such as “(e.g., tunnel handling or cupping)”

18 – A better rationale would be to mention that it’s known that at least some labs are known not to use the methods, but the extent and reasons are unknown.

34 – If space, add a conclusion statement or two of overarching conclusions of the research, additions, or next steps.

36 – If possible, add keyword of “human-animal interaction” for broader reach

Introduction

38 – Good intro paragraph

45 – 65 – Really good information and summary of non-aversive handling methods.

67 - A better rationale would be to mention that it’s known that at least some labs are known not to use the methods, but the extent and reasons are unknown. You can use terms such as “suspected” or “anecdotally” low levels/barriers.

Methods

78 – Was this protocol reviewed by a human subjects research board? If so, please give the protocol number in the methods? If not, why did you not seek review?

80 - What were your inclusion criteria/how did you screen participants?

105 – Some of the details in the analysis section such as whether a list was used, or the number of options given for a response do not seem relate to analysis and rather are more descriptive about the survey. Move these up.

113 – State that these groupings were only used for analysis, but that all statistics were reported. I initially thought you were not going to give all the percentages per category.

129 – How many respondents provided qualitative comments? Also were percentages taken out of all of the participants or all of the respondents to the question? (The former I think would be more accurate.)

Analysis – In this section you should provide indication/rationale for why you will be splitting results based on job role.

Results

148 – Give rationale for why you’re splitting up these statistics by job role so it is clear for the reader.

167-175 – Clear reporting/summary in text.

230-235 – This figure could be summarized better. Focus more on the overarching conclusions rather than the differences by job role.

247 – Better to list the % of respondents. Otherwise it’s a bit misleading.

256 – If you repeat the N, consider also repeated the %.

261-264 – It’s unclear whether the same or different respondents indicated that mice are often already residing in the tunnel. If it’s different respondents then please cut this. Use the discussion section to provide additional information & discuss conflicting opinions.

271 – 273 – Save the interpretation of this response for the discussion.

Discussion

322 – The first sentence is a bit hard to digest. Be sure to indicate that “after reading information” (or state otherwise) they thought they’d be beneficial and that the “currently” certain percentages use each methods.

334 – To me, it seems like even if the maximum amount use non-aversive methods (61%), there is still a need for more implementation of them. Even in the best case scenario, more work is needed as it’s barely more than half. And in the worst case scenario it’s only 18%. Therefore I think this survey highlights the need for greater implementation.

339 – remove the comma before the “and”

343-347 – This is good interpretation, but could be stated more simply. I suggest something similar to “These results may indicate that the benefits of non-aversive handling to scientific outcomes are either less important or less convincing as compared to welfare outcomes.” Then summarize the supporting data. In 346 just say “participants” vs “researchers and animal care staff”.

347 – Sub “additionally” for “whereas” and cut “the number of participants that were unconvinced differed very little between welfare & scientific outcomes.” I think this will help with comprehension and clarity of this point, which I think is a good one, but is getting lost in the sentence structure. I don’t think that the number that were unconvinced speaks to participants ability to judge the validity of data as they are specifically saying that were not convinced.

362-367 – Awesome insight!

369 – It is important to indicate your sample size here and the % of participants that actually replied to this question. I’m not sure if you actually state that in your results, but it is important for how far to draw your conclusions.

369-370 – Is the concern it would be slower part of the thematic analysis or was that a closed response? If so, the topic sentence of this paragraph is a bit misleading. Also then sentence 375-376 is confusing as well and seems to repeat 370-372.

385-386 – good conclusion

403 – why is this sentence bold?

471 – A limitation section should be added. Key point is to indicate is that this was a convenience sample.

471 – A final conclusion paragraph should be added indicating the contribution of this experiment to the field & big over-arching findings.

473 -491 – I like this recommendation section

Tables & Figures

Table 1 – The inclusion of the % per category aware of non-aversive handling seems odd to me and the rationale of it is not well explained. Also with so few respondents in some categories it may be slightly misleading (e.g., saying that 100% of respondents from India were aware of non-aversive handling when there was only 1 respondent from India).

Table 2 – Similar to the comment above, but with the gender split & % female. This doesn’t seem to answer a study aim and provides excessive information.

Table 3 – Same comment as above. You also don’t mention splitting by sex in text. This is extraneous information. Also I would not provide the specifics under “tail and non-aversive.” This could also be nice as a visual chart instead. Easier to absorb as a reader.

Table 4 – It is a bit confusing to have both closed-choice and thematically coded text in the same table (especially since Table 5 specifically indicates thematically coded responses for the other question). Please remove the thematically coded text and instead only include that in text or instead in a different part of the table. Otherwise it looks like all options were available to respondents and few chose the other ones. If you choose to include as a sub table, order responses from most common (mouse already in tunnel) to least common (personal well-being).

Table 5 – In the table description make sure to indicate that these responses were thematically coded. Technically these are the “themes” of responses, not the reasons themselves.

Figure 1 – Provide a bar & “*” between the comparisons that were significantly different.

Figure 1 & 2 – Use a gradient color scale to more intuitively indicate stronger/more often responses with darker colors and weaker/less often responses with light color (e.g., < 1 month could be white and every working day could be black or a dark shade of blue or whatever color you like). Also abbreviate more than as “>” and less than as “<” Then the legend will likely fit on one line and be easier to read/comprehend.

Figure 2 – Right now the tail & non-aversive and non-aversive bars really jump out at the reader. Is that what you want to highlight? It may be better to specifically choose a darker color for what you want to highlight (non-aversive methods) and then let the others be similar colors since perhaps they are similar? It depends on the story you want to tell.

Figure 3 – Bold highlighting does not come across. Instead consider an asterix only for the significantly difference methods. There appears to be a ¼ after “time required for retraining.” As English readers will read from top to bottom, order the options from most to least common from top to bottom. I would also consider just making this into a table, as it is a bit difficult to read as a figure. Left justification of the response text may help.

Figure 4 – These graphs are fairly difficult to absorb as a viewer. Use color intentionally to indicate responses. The not convinced & not very convinced bars should be similar colors since they indicate similar responses. No opinion should be colored grey or white since it is neutral. Mildly & Very should be a different color. See this article for another way & advice for creating graphs for likert responses - https://stephanieevergreen.com/aggregated-stacked-bars/

Figure 4C - The differences in views part of the table is confusing and seems to repeat the above tables. I would cut this and instead summarize in text. I’m not sure this granulation of responses is necessary/useful. Or somehow change the graph, but it’s fairly difficult to absorb on it’s own.

Reviewer #2: I was very pleased to see a manuscript based on feedback from researchers and animal care staff on the use of different handling methods, and I only have a couple of comments:

Comment 1:

Effects of tail handling on anxiety are widely reported and have been replicated by many research groups. My main concern is that these effects have only been found after extensive daily handling and while this type of handling may be relevant for some types of research (e.g. habituating the animals to handling before behavioural testing), the results may not apply to routine husbandry practices with handling of mice once or twice per week during cage changes and other routine procedures.

Therefore, this manuscript is a good opportunity to provide a more balanced view on impacts of handling method on mouse welfare and data quality and I would like to see a more critical view on the existing literature on effects of handling method. In the Introduction, the authors list the studies showing impacts of tail handling on anxiety measures, anhedonia etc, but these effects are strain dependent (Gouveia and Hurst, Reducing mouse anxiety during handling: effect of experience with handling tunnels, Hurst and West, Taming anxiety in laboratory mice), sex dependent (Gouveia and Hurst, Improving the practicality of using non-aversive handling methods to reduce background stress and anxiety in laboratory mice), and contradictory findings have been found (Nakamura et al).

In Lines 59-60, the authors say that "Furthermore, handling method can impact physiological indices" and quote two references (Ghosal et al and Ono et al). This is a very vague statement, and the authors do not specify the results of the referenced studies in more detail, which would clarify that handling method impacts some, but not all physiological indices (Ghosal). Ono et al found that handling (regardless of method) has an effect on renal disease model phenotype, they even report higher blood corticosterone levels in tunnel handled mice (again, only one strain and one sex).

Overall, it would be useful if the effects of non-aversive handling would be put in perspective and more critically assessed (Line 64-65). This is also echoed by the responses of participants in the study – page 18, Lines 293-312

Comment 2:

The results and discussion could be split into two main points:

1. issues dealing with practicality, cost and training of tunnel handling - which can be addressed by outreach, training etc and

2. The need for further evidence in support of using non-aversive handling method. The fact that 30-40% of respondents said they are mildly convinced that non-aversive handling methods improve animal welfare and scientific outcomes (Fig 4), the authors could place more focus on the need for further studies facilitating a more accurate (and real life scenario) assessment of the effects of handling procedures on the data quality and wellbeing of mice. This is addressed in the discussion (Page 23, Lines 428-443), but it seems that the authors cite the literature in support of tunnel handling (and they could be more critical in evaluating it, see Comment 1).

6. PLOS authors have the option to publish the peer review history of their article (what does this mean?). If published, this will include your full peer review and any attached files.

Reviewer #1: Yes: Megan R. LaFollette

Reviewer #2: No

---

## [Author Response · Author response to Decision Letter 0]

21 Mar 2020

Dear Editor(s),

Thank you for the opportunity to submit a revised version of our manuscript (PONE-D-19-32020), “Identifying obstacles preventing the uptake of tunnel handling methods for laboratory mice: An international thematic survey”. We were pleased to see that you and the two reviewers judged the manuscript to be timely, well written and well designed. 

We are very grateful to you and the reviewers for their advice; they have provided insightful comments and questions, and in responding to them, we believe the manuscript is very much improved.

We have now carefully considered the questions and suggestions from the reviewers’ and have produced a detailed description of how we have improved our manuscript in response to them. Our responses to the reviewers are in bold below their comments. In the edited version of the manuscript, changes are highlighted in green.

Sincerely, 

Lindsay Henderson

Reviewer #1:

Solid manuscript and experiment overall. Well written, well designed, and timely. My comments below are written with the line number they reference proceeding.

Abstract

14 – If there is space include that handling methods can also influence physiological data. Also the sentence would be better written as a list such as “influence behavior, potentially reduce replicability…, and compromise mouse welfare.” The fact that the method used influences performance in behavioral test is not the reason that mouse welfare is compromised.

 This sentence has been changed to “Handling of laboratory mice is essential for experiments and husbandry, but handling can increase anxiety in mice, compromising their welfare and potentially reducing replicability between studies”, on lines 14-15. We have not referenced physiological data as this is limited, and in accordance with the comments of Reviewer 2.

16 – If space, include brief definition of non-aversive handling such as “(e.g., tunnel handling or cupping)”

This has been added on line 16.

18 – A better rationale would be to mention that it’s known that at least some labs are known not to use the methods, but the extent and reasons are unknown.

This has been added on line 19, “some labs continue to use tail handling for routine husbandry”.

34 – If space, add a conclusion statement or two of overarching conclusions of the research, additions, or next steps.

 This has been added on lines 35-37.

36 – If possible, add keyword of “human-animal interaction” for broader reach

This has been added to the Keywords.

Introduction

38 – Good intro paragraph

45 – 65 – Really good information and summary of non-aversive handling methods.

 Thank you.

67 - A better rationale would be to mention that it’s known that at least some labs are known not to use the methods, but the extent and reasons are unknown. You can use terms such as “suspected” or “anecdotally” low levels/barriers.

Thank you for your comment. We have added this on lines 84-86, “tail handling continues to be used for routine handling in some laboratories, and the extent to which non-aversive methods are being routinely used is unknown”.

Methods

78 – Was this protocol reviewed by a human subjects research board? If so, please give the protocol number in the methods? If not, why did you not seek review?

Yes, the anonymous questionnaire was reviewed by the Newcastle University Ethics committee. See line 103.

80 - What were your inclusion criteria/how did you screen participants?

We did not screen participants; we have added a sentence to clarify this on lines 101-102. “All participants that completed the survey were included in analysis, participants were not screened, and no inclusion or exclusion criteria were used.”

105 – Some of the details in the analysis section such as whether a list was used, or the number of options given for a response do not seem to relate to the analysis and rather are more descriptive about the survey. Move these up.

We have not moved text from this section but based on your comments we have renamed the section to, “Data processing and analysis”, to more accurately describe its’ content.

113 – State that these groupings were only used for analysis, but that all statistics were reported. I initially thought you were not going to give all the percentages per category.

We have now stated “Where appropriate job roles were used for analysis and descriptive statistics.” on lines 147-148.

129 – How many respondents provided qualitative comments? Also were percentages taken out of all of the participants or all of the respondents to the question? (The former I think would be more accurate.)

 The percentages that were reported were the proportion of the total number of respondents that provided a response to the question that could be coded into a theme. The question being “What would be required for you to consider using tunnel handling routinely?”. Not the proportion of the total number of participants that completed the survey. We take your point that this may not give an accurate view of the results, so we have made changes to the Methods and Results sections to reflect this.

 249 participants provided qualitative comments, see lines 165-169 in the methods section. Of these comments, 208 provided responses to the question that could be categorized into themes. For example, some comments could not be coded into themes as they stated responses like; “I will now start using tunnel handling” or “I already use tunnel handling or cup handling routinely”. 

Importantly, this question specifically targeted respondents that were not routinely using tunnel handling. So, we expected responses only from a sub-sample of respondents, i.e. those not routinely using tunnel handling. Because of this we do not think only reporting the % as a proportion of the total number of survey participants is more accurate. Instead we have added further information to explain this in the methods section on lines 165-169;

…“Not all respondents provided qualitative comments for the question, “What would be required for you to consider using tunnel handling routinely?”, that offered a free-form response, as the question was optional (N = 249). As this question specifically targeted respondents that were not routinely using tunnel handling, responses were gathered from a sub-sample of respondents.”

Also, in the Results section we now provide a more in-depth description of the sub-sample of respondents that provided the responses used in the thematic analysis, in relation to the total number of survey participants. See lines 294-304 of the Results section. We also explain the percentages reported in the thematic analysis section.

“As this question specifically targeted respondents that were not routinely using tunnel handling, we expected responses from a sub-sample of respondents. 249 out of 390 total survey participants provided free-text responses (64%), 208 of those could be coded and used in the thematic analysis (53% of total survey participants). 43% of total survey participants were potentially not routinely using tunnel handling, as they reported using a combination of tail and non-aversive methods, and 39% of total survey participants reported not using any form of non-aversive methods (82% of total survey participants, see Table 3). Therefore, a substantial proportion of respondents, that may not have been using tunnel handling routinely, provided qualitative responses to the question. Percentages reported below are as a proportion of the total number of responses that could be thematically coded (N = 208).”

Analysis

In this section you should provide indication/rationale for why you will be splitting results based on job role.

Please see lines 139-148.

Results

148 – Give rationale for why you’re splitting up these statistics by job role so it is clear for the reader.

We have added these details in the methods section on lines 139-148.

167-175 – Clear reporting/summary in text.

230-235 – This figure could be summarized better. Focus more on the overarching conclusions rather than the differences by job role.

Figure 3 (was Fig. 4) legend has been re-worded. Please see response to your comments for Tables & Figures.

247 – Better to list the % of respondents. Otherwise it’s a bit misleading.

Thank you for your comment. As outlined above this is the % of responses to the question. In this section we have not changed this. We think this is a more accurate representation of participants who would have answered this question, i.e. those that were not routinely using tunnel handling. However, we have explained our rationale for this further in the Methods and Results sections of the manuscript in response to your other comments. We have also shown the proportion of responses as a percentage of total survey participants. Please also see our responses to your comments above.

256 – If you repeat the N, consider also repeated the %.

We have removed the Ns here as we have added more details about the sample size on lines 294-304 of the Results section.

261-264 – It’s unclear whether the same or different respondents indicated that mice are often already residing in the tunnel. If it’s different respondents, then please cut this. Use the discussion section to provide additional information & discuss conflicting opinions.

Thank you for your comment. Yes, it was different respondents that said tunnel handling was faster than tail handling. We have added “other” on line 297 to clarify this. As this is a description of the comments provided we have not removed this sentence from the results section. We discuss that fact that there are conflicting opinions in the discussion on lines 434-455. 

271 – 273 – Save the interpretation of this response for the discussion.

These sentences have been removed, see line 322. This is now only examined in the Discussion, please see lines 434-455.

Discussion

322 – The first sentence is a bit hard to digest. Be sure to indicate that “after reading information” (or state otherwise) they thought they’d be beneficial and that the “currently” certain percentages use each method.

This paragraph has now been re-worded to reflect your comment. We have not specifically added “after reading information” because information was offered to each participant and they were encouraged to read the information but there was no way to enforce this. Therefore, we cannot state that they had read the information through completing the survey. We assume they had read about non-aversive handling before and/or during the survey. Especially as the majority of participants had heard of non-aversive handling before completing the survey. Rather than saying currently we have changed the language to say “reported using”, i.e. those were the methods they used when they completed the survey, the methods they are now using may have changed. Please see lines, 382-402.

334 – To me, it seems like even if the maximum amount use non-aversive methods (61%), there is still a need for more implementation of them. Even in the best-case scenario, more work is needed as it’s barely more than half. And in the worst-case scenario it’s only 18%. Therefore, I think this survey highlights the need for greater implementation.

We have added this to lines, 399-401.

339 – remove the comma before the “and”

Comma has been removed.

343-347 – This is good interpretation but could be stated more simply. I suggest something similar to “These results may indicate that the benefits of non-aversive handling to scientific outcomes are either less important or less convincing as compared to welfare outcomes.” Then summarize the supporting data. 

We have changed this sentence, now on line 408-410.

In 346 just say “participants” vs “researchers and animal care staff”.

We have now said participants, see line 412.

347 – Sub “additionally” for “whereas” and cut “the number of participants that were unconvinced differed very little between welfare & scientific outcomes.” I think this will help with comprehension and clarity of this point, which I think is a good one, but is getting lost in the sentence structure. I don’t think that the number that were unconvinced speaks to participants’ ability to judge the validity of data as they are specifically saying that were not convinced.

We have changed “whereas” to “additionally” on line 413. We have also split up the sentences so that we are stating it is the greater selection of “no opinion” for scientific outcomes that may reflect concerns by participants of their ability to judge the validity of data. See lines 414-416.

362-367 – Awesome insight!

369 – It is important to indicate your sample size here and the % of participants that actually replied to this question. I’m not sure if you actually state that in your results, but it is important for how far to draw your conclusions.

Thank you for your insight, we have stated this more clearly in the results section in response to your comments, so that the reader can judge how much weight to give our conclusions. As we have outlined above, we have given a clearer description of the proportion of total respondents that responded to this question. Also see our response to your comments above. See changes to Methods and Results.

369-370 – Is the concern it would be slower part of the thematic analysis or was that a closed response? If so, the topic sentence of this paragraph is a bit misleading. Also, then sentence 375-376 is confusing as well and seems to repeat 370-372.

Thank you for your comment, we have now removed this sentence. Time constraints were one of the most commonly chosen reasons for not routinely using tunnel handling. They were also the most common issue highlighted by the thematic analysis. 

385-386 – good conclusion

 Thank you.

403 – Why is this sentence bold?

This sentence was not in bold in our version, this may have been a pdf conversion issue. It is no longer in bold.

471 – A limitation section should be added. Key point is to indicate is that this was a convenience sample.

We have not explicitly stated that this survey was a convenience sample, but we have clearly outlined our sample size and methods for recruiting participants, so that readers can assess the confidence they should place on our results. We draw attention to other limitations of the survey throughout.

471 – A final conclusion paragraph should be added indicating the contribution of this experiment to the field & big over-arching findings.

We have now added a final paragraph at the end of the Discussion.

473 -491 – I like this recommendation section

Thank you.

Tables & Figures

Table 1 – The inclusion of the % per category aware of non-aversive handling seems odd to me and the rationale of it is not well explained. Also with so few respondents in some categories it may be slightly misleading (e.g., saying that 100% of respondents from India were aware of non-aversive handling when there was only 1 respondent from India).

We agree with your comment and have now removed these data from the table. A written description of the % of respondents that had heard of tunnel handling is still in the main text.

Table 2 – Similar to the comment above, but with the gender split & % female. This doesn’t seem to answer a study aim and provides excessive information.

We agree with your comment and have now removed these data from the table.

Table 3 – Same comment as above. You also don’t mention splitting by sex in text. This is extraneous information. Also I would not provide the specifics under “tail and non-aversive.” This could also be nice as a visual chart instead. Easier to absorb as a reader.

We agree with your comment and have now removed the gender split from the table. However, we have left the specifics of the methods used in the table. To improve clarity, we have ordered responses from most common to least common.

Table 4 – It is a bit confusing to have both closed-choice and thematically coded text in the same table (especially since Table 5 specifically indicates thematically coded responses for the other question). Please remove the thematically coded text and instead only include that in text or instead in a different part of the table. Otherwise it looks like all options were available to respondents and few chose the other ones. If you choose to include as a sub table, order responses from most common (mouse already in tunnel) to least common (personal well-being).

We agree with your comment and we have split the responses within the table. We now clearly explain in the table that some responses when chosen from answer options provided, and some were collected from the other option. We have also ordered responses from most common to least common as suggested.

Table 5 – In the table description make sure to indicate that these responses were thematically coded. Technically these are the “themes” of responses, not the reasons themselves.

We have added this to the legend of Table 5.

Figure 1 – Provide a bar & “*” between the comparisons that were significantly different.

Figure 1 & 2 – Use a gradient color scale to more intuitively indicate stronger/more often responses with darker colors and weaker/less often responses with light color (e.g., < 1 month could be white and every working day could be black or a dark shade of blue or whatever color you like). Also abbreviate more than as “>” and less than as “<” Then the legend will likely fit on one line and be easier to read/comprehend.

Figure 2 – Right now the tail & non-aversive and non-aversive bars really jump out at the reader. Is that what you want to highlight? It may be better to specifically choose a darker color for what you want to highlight (non-aversive methods) and then let the others be similar colors since perhaps they are similar? It depends on the story you want to tell.

Figure 3 – Bold highlighting does not come across. Instead consider an asterix only for the significantly difference methods. There appears to be a ¼ after “time required for retraining.” As English readers will read from top to bottom, order the options from most to least common from top to bottom. I would also consider just making this into a table, as it is a bit difficult to read as a figure. Left justification of the response text may help.

Figure 4 – These graphs are fairly difficult to absorb as a viewer. Use color intentionally to indicate responses. The not convinced & not very convinced bars should be similar colors since they indicate similar responses. No opinion should be colored grey or white since it is neutral. Mildly & Very should be a different color. See this article for another way & advice for creating graphs for likely responses - https://stephanieevergreen.com/aggregated-stacked-bars/

Figure 4C - The differences in views part of the table is confusing and seems to repeat the above tables. I would cut this and instead summarize in text. I’m not sure this granulation of responses is necessary/useful. Or somehow change the graph, but it’s fairly difficult to absorb on it’s own.

Thank you for your comments we have made the changes you suggested and believe it has improved our figures. We have used gradient colours for Figs 1 & 2, the colours we have used come from colorbrewer

(https://colorbrewer2.org/#type=sequential&scheme=YlGnBu&n=3), which provides colour gradient and divergent colour schemes that are colour blind friendly and optimize colour differences for perception. We have added “**” on Fig 2. Also, Fig 2 is now ordered based on the most commonly used handling method, and we have added a sentence describing the figure to the legend. For Fig. 2 we have also added a reference to a supplementary figure (3) that compared handling methods between the UK and the other countries represented.

 We have replaced Fig 3 with a Table (Table 6). For Fig 4 (now Fig 3) we have used colours as suggested and removed 4c. 

Reviewer #2: I was very pleased to see a manuscript based on feedback from researchers and animal care staff on the use of different handling methods, and I only have a couple of comments:

Comment 1:

1.1 Effects of tail handling on anxiety are widely reported and have been replicated by many research groups. My main concern is that these effects have only been found after extensive daily handling and while this type of handling may be relevant for some types of research (e.g. habituating the animals to handling before behavioural testing), the results may not apply to routine husbandry practices with handling of mice once or twice per week during cage changes and other routine procedures.

This is an important point and we have outlined new evidence investigating this issue, now on lines 76-79 in the Introduction, we have also referenced two more recent papers (included our own work) that have shown that weekly or fortnightly handling during cage cleaning continues to influence behaviour. Also, in the Discussion, lines 511-515.

1.2 Therefore, this manuscript is a good opportunity to provide a more balanced view on impacts of handling method on mouse welfare and data quality and I would like to see a more critical view on the existing literature on effects of handling method. In the Introduction, the authors list the studies showing impacts of tail handling on anxiety measures, anhedonia etc, but these effects are strain dependent (Gouveia and Hurst, Reducing mouse anxiety during handling: effect of experience with handling tunnels, Hurst and West, Taming anxiety in laboratory mice), sex dependent (Gouveia and Hurst, Improving the practicality of using non-aversive handling methods to reduce background stress and anxiety in laboratory mice), and contradictory findings have been found (Nakamura et al).

We highlight strain dependence, please see lines 65, 462-465 and 476. We suggest that while the results of previous studies are not identical (we have highlighted this in the article see lines 476, 568), they are broadly in agreement, i.e. the vast majority of effects of handling method upon behaviour are in the same direction. Therefore, we feel itemizing each result highlighted in your comment in the manuscript is not necessary to give an overall description of current research. We also restrict our statement to effects of handling upon interaction with a handler that is consistent across studies. Please see an outline of results from the studies you mention below to explain our position;

Strain dependence

Gouveia and Hurst, Reducing mouse anxiety during handling: effect of experience with handling tunnels

Comparing shared tunnels C57 and ICR mice. ICR showed significant difference between handling methods after one session. Whereas, for C57 mice this was only the case after 9 sessions. However, both strains reacted similarly to home tunnels.

• This is highlighted on lines 462-465.

Hurst and West, Taming anxiety in laboratory mice.

For Voluntary Interaction tests, strains differ in effects sizes but show the same overall direction of effects caused by handling methods. 

For the EPM, dark phase testing, tail handled BALB/c, C57 and ICR show significantly more protected stretch attend poses than tunnel handled mice. Tail handled C57 and ICR showed significantly fewer open arm entries, but C57s did not. Tail handled BALB/c (Female only) and ICR spent less time on the open arms, but C57s did not.

For the EPM, light phase testing, tail handled BALB/c and C57s showed significantly more protected stretch attend poses. Tail handled C57 showed significantly fewer open arm entries, but BALB/c did not. Also, tail handled BALB/c showed significantly more entries into open arm than tunnel handled.

Sex dependence

Gouveia and Hurst, Improving the practicality of using non-aversive handling methods to reduce background stress and anxiety in laboratory mice

There were significant sex differences in behaviour, but all in the same direction, tunnel versus tail. The exception was that there was no difference for male mice between tail and tunnel handling for time spent on open arms of the EPM.

Contradictory findings have been found

Nakamura et al., Tunnel use facilitates handling of ICR mice and decreases experimental variation.

For the EPM, tail handled mice spent greater % time on open arms than tunnel handled mice. However, other results were in agreement with tunnel handled mice being less anxious. Tail handled mice showed fewer entries onto all arms (had lower overall activity) and specifically onto the open arms.

For Voluntary Interaction tests, significantly higher interaction for tunnel handled throughout the experiment. Amount of interaction did decline after oral administration particularly for tunnel handled mice. However, the difference in behaviour continued to be significantly different between handling methods, with tunnel handled mice showing higher levels of interaction.

Tunnel handled mice were easier to handle throughout and had lower levels of urination and defecation.

CVs for all items except % time in open arms were greater in the tail handling group compared to the tunnel handling group.

1.3 In Lines 59-60, the authors say that "Furthermore, handling method can impact physiological indices" and quote two references (Ghosal et al and Ono et al). This is a very vague statement, and the authors do not specify the results of the referenced studies in more detail, which would clarify that handling method impacts some, but not all physiological indices (Ghosal). Ono et al found that handling (regardless of method) has an effect on renal disease model phenotype, they even report higher blood corticosterone levels in tunnel handled mice (again, only one strain and one sex). Overall, it would be useful if the effects of non-aversive handling would be put in perspective and more critically assessed (Line 64-65). This is also echoed by the responses of participants in the study – page 18, Lines 293-312.

Thank you for your comment. We have changed the language to reflect your comment, we now state “To date evidence regarding the impact of handling method upon physiological indices is limited. A single study has shown that handling method can influence glucose metabolism, and there are inconsistent results regarding the influence of handling methods upon plasma corticosterone levels from two studies.”. Please see lines 70-74. We have also elaborated on this point in the Discussion, to highlight the limitations of physiological data, and need for further research. Please see lines 547-559.

We have outlined the Ono et al. study in the Discussion. The statement in Ono et al. (In both male and female mice, handled mice appeared to show a more severe lesion grade level than the control mice regardless of the handling method) does not have a statistical test. The only significant statistical test reported is the difference between female tail handled and female control (unhandled) mice. We report this finding and that tunnel and cup handling mice scores did not differ significantly from either tail handled or control mice on lines 533-535. 

Comment 2:

The results and discussion could be split into two main points:

1. issues dealing with practicality, cost and training of tunnel handling - which can be addressed by outreach, training etc and

2. The need for further evidence in support of using non-aversive handling method. The fact that 30-40% of respondents said they are mildly convinced that non-aversive handling methods improve animal welfare and scientific outcomes (Fig 4), the authors could place more focus on the need for further studies facilitating a more accurate (and real life scenario) assessment of the effects of handling procedures on the data quality and wellbeing of mice. This is addressed in the discussion (Page 23, Lines 428-443), but it seems that the authors cite the literature in support of tunnel handling (and they could be more critical in evaluating it, see Comment 1).

 As there are overlapping issues, we have not divided the results and discussion into two sections. However, we have moved the paragraph discussing the issues with perceived incompatibility of tunnels with experimental apparatus up, so that it follows the paragraph about lack of tunnels and biosecurity issues.

---

## [Editor Report · Decision Letter 1]

25 Mar 2020

Identifying obstacles preventing the uptake of tunnel handling methods for laboratory mice: An international thematic survey

PONE-D-19-32020R1

Dear Dr. Henderson,

We are pleased to inform you that your manuscript has been judged scientifically suitable for publication and will be formally accepted for publication once it complies with all outstanding technical requirements.

With kind regards,

Kathleen R. Pritchett-Corning, D.V.M.

Academic Editor

PLOS ONE

Additional Editor Comments (optional):

Thanks for your patience. Your attention to reviewer comments has greatly improved the MS.
---

## [Editor Report · Acceptance letter]

31 Mar 2020

PONE-D-19-32020R1 

Identifying obstacles preventing the uptake of tunnel handling methods for laboratory mice: An international thematic survey 

Dear Dr. Henderson:

I am pleased to inform you that your manuscript has been deemed suitable for publication in PLOS ONE. Congratulations! Your manuscript is now with our production department. 

With kind regards,

on behalf of

Dr. Kathleen R. Pritchett-Corning 

Academic Editor

PLOS ONE